# Rab8, Rab11, and Rab35 coordinate lumen and cilia formation during zebrafish left-right organizer development

**Abrar A. Aljiboury** [1,2☯], **Eric Ingram** [1,2☯¤a], **Nikhila Krishnan** [1,2☯¤b], **Favour Ononiwu** [1,2☯], **Debadrita Pal** [1,2☯¤c], **Julie Manikas** [1¤d], **Christopher Taveras** [1], **Nicole A. Hall** [1¤d], **Jonah Da Silva** [1], **Judy Freshour** [1], **Heidi Hehnly** [1,2] *

1 Biology Department, Syracuse University, Syracuse, New York, United States of America, 2 BioInspired Institute, Syracuse University, Syracuse, New York, United States of America

☯ These authors contributed equally to this work.
¤a Current address: Thermo Fisher, Madison, Wisconsin United States of America
¤b Current address: Department of Biology, Brandeis University, Waltham, Massachusetts, United States of America
¤c Current address: Sanofi Global, Cambridge, Massachusetts, United States of America
¤d Current address: Department of Cell Biology, NYU Grossman School of Medicine, New York, New York, United States of America
* hhehnly@syr.edu

**Data Availability Statement:** All relevant data are within the manuscript and its supporting information files.

## Abstract

An essential process during *Danio rerio's* left-right organizer (Kupffer's Vesicle, KV) formation is the formation of a motile cilium by developing KV cells which extends into the KV lumen. Beating of motile cilia within the KV lumen directs fluid flow to establish the embryo's left-right axis. However, the timepoint at which KV cells start to form cilia and how cilia formation is coordinated with KV lumen formation have not been examined. We identified that nascent KV cells form cilia at their centrosomes at random intracellular positions that then move towards a forming apical membrane containing cystic fibrosis transmembrane conductance regulator (CFTR). Using optogenetic clustering approaches, we found that Rab35 positive membranes recruit Rab11 to modulate CFTR delivery to the apical membrane, which is required for lumen opening, and subsequent cilia extension into the lumen. Once the intracellular cilia reach the CFTR positive apical membrane, Arl13b-positive cilia extend and elongate in a Rab8 dependent manner into the forming lumen once the lumen reaches an area of 300 µm². These studies demonstrate the need to acutely coordinate Rab8, Rab11, and Rab35-mediated membrane trafficking events to ensure appropriate timing in lumen and cilia formation during KV development.

## Author summary

Cilia are microtubule-based structures that extend from the cell surface to sense and transmit extracellular cues to the cell body. Defective cilia result in a series of conditions known as ciliopathies. Although the mechanisms of cilia formation have been elucidated *in vitro*, how a cilium is made *in vivo* during embryogenesis is less studied. We take

**Funding:** This work was supported by National Institutes of Health grants R01GM127621 (H.H.) and R01GM130874 (H.H.). The funders had no role in study design, data collection and analysis, decision to publish, or preparation of the manuscript.

**Competing interests:** The authors have declared that no competing interests exist.

advantage of the Zebrafish's organ of asymmetry, also known as the Kupffer's Vesicle (KV), as a model to study cilia formation during vertebrate embryogenesis. In the KV, cilia and lumen formation occur concurrently, making it a great model to study how these processes are coordinated. Here we show that KV precursor cells (dorsal forerunner cells) form cilia within the cell volume before extending into the lumen of the KV. To determine the molecular mechanisms of cilia formation during KV development, we focus on three Rab GTPases, Rab11, Rab8 and Rab35, that have been implicated in cilia or lumen formation independently *in vitro*. We use optogenetic clustering in parallel with morpholino depletion approaches to show that these three Rab GTPases coordinate specific events that govern cilia and lumen formation during KV development.

## Introduction

A fundamental question in cell biology is how a cilium is made during tissue formation. A primary or motile cilium is a microtubule-based structure that extends from the surface of a cell and can sense extracellular cues to transmit to the cell body. Defects in cilia formation can lead to numerous disease states collectively known as ciliopathies [1,2]. Foundational studies identified two distinct pathways for ciliogenesis *in vivo* using tissues from chicks and rats [3]. One mechanism for ciliogenesis which we refer to as extracellular, was found in lung cells where the centrosome first docks to the plasma membrane followed by growth of the ciliary axoneme into the extracellular space [4]. The second mechanism, which we refer to as intracellular, was identified in smooth muscle cells and fibroblasts where the centrosome forms a cilium first within a ciliary vesicle in the cell cytosol before docking to the plasma membrane [3]. These studies raise the possibility that different ciliated tissues construct their cilia differentially due to the nature of how a tissue develops. This presents an important hypothesis that variations in cilia formation mechanisms may occur *in vivo* during specific types of tissue morphogenesis.

Here we examine cilia formation during *Danio rerio* (zebrafish) organ of asymmetry (Kupffer's Vesicle, KV) development. The KV is required to place visceral and abdominal organs with respect to the two main body axes of the animal [5]. KV formation begins from a subset of dorsal enveloping layer (EVL) cells called dorsal forerunner cells (DFCs) [6,7]. These DFCs are precursors of the KV that situate posteriorly to the notochord [6,8,9]. The number of DFCs range from 10–50 cells per embryo that can expand into >100 cells that make up the fully functional KV [10,11]. Early studies reported that these DFCs present as mesenchymal-like and are migratory. They lack clear apical/basal polarity until KV cells establish into rosette-like structures [9]. Apical polarity establishment of atypical Protein Kinase C (aPKC), at least in part, coincides with cystic fibrosis transmembrane conductance regulator (CFTR) accumulation at apical sites, which is a requirement for lumen expansion [9,12]. KV cell rosette-like structures can either form as multiple cells congressing to make a single rosette or cells assembling multiple rosettes, which then transition to a single rosette-like structure. The rosette center is the site where a fluid-filled lumen forms and KV cells will then extend their cilia into [12]. Once KV cilia are formed, they beat to direct fluid flow essential for the establishment of the embryo's left-right axis [13]. While much is known about KV post-lumen formation [5,14–17], little is known about the spatial and temporal mechanisms that regulate cilia formation during KV development.

*In vitro* cell culture assays have been used to identify regulators of lumen establishment or ciliogenesis and have identified 3 Rab GTPases that have been implicated in both processes—Rab11, Rab8, and Rab35 [18–24]. While select Rab GTPases have been extensively studied,

most of them have not been assigned to a detailed function or localization pattern during early embryonic vertebrate development. Rab GTPases comprises approximately 60 genes in vertebrates, with each Rab localizing to specific intracellular membrane compartments in their GTP-bound (active) form. These active Rabs then bind to effector proteins to aid in various steps in membrane trafficking, some of which will facilitate cilia, polarity, and/or lumen formation [25,26]. In our studies herein, we focus on Rab8, Rab11, and Rab35 that have been linked to lumen or cilia formation independently, but have not been tested in experimental setups where these events occur concurrently [18,19,21,27,28]. Specifically, we investigate the role of these three Rab GTPases in KV development using a combination of depletion and optogenetic clustering approaches. Through these approaches we have identified conserved and unique roles for Rab8, Rab11 and Rab35 in coordinating KV lumen and cilia formation. Our findings were surprising in that Rab8 did not seem to affect lumen or cilia formation to a similar extent that it does in mammalian cell culture [18–21]. In mammalian cells, Rab8 and Rab11 work together in a GTPase cascade that is required for both cilia and lumen formation. However, in the KV, lumen formation seems to be dependent on a coordination between Rab35 and Rab11, while Rab8 is dispensable for this process. These results suggest that specific cell types during potentially different developmental processes may have varying dependencies on Rab GTPases that can be identified using developmental model systems such as zebrafish.

## Results

### Kupffer's Vesicle (KV) cilia form prior to KV lumen formation using an intracellular pathway

While studies have demonstrated that the majority of KV cells have cilia when KV lumen formation has already occurred, KV cilia formation has not been carefully characterized. To test when during KV development cilia form, we fixed embryos at the bud stage, 1 Somite Stage (SS), and 6 SS. At these stages, the KV is primarily organized as a pre-rosette (Bud), rosette (1 SS) and lumen (6 SS) (Fig 1A and 1B). Embryos were immunostained for cilia using an antibody against acetylated tubulin (ac-tubulin) and centrosomes using an antibody against γ-tubulin (Fig 1B). In addition, actin was stained to mark cell boundaries and denote the rosette center at the rosette stage (Figs 1B and S1A). Strikingly, we identified that a significant population of KV cells started to form cilia at the centrosome within the cell body before a KV lumen has formed (Fig 1B). During the pre-rosette stage, 33.25±3.33% of KV cells already had cilia; that increased to 48.06±5.94% at the rosette stage and averaged at 64.82±5.27% early on during lumen formation (Fig 1B and 1D). These studies suggested that KV cells were forming cilia before they had an extracellular space (KV Lumen), and that KV lumen formation correlated with a significant increase in KV cells having cilia.

We next examined when cilia extend into the forming KV lumen. To do this we employed two strategies: we imaged live Sox17:GFP-CAAX embryos that ectopically expressed the cilia marker Arl13b-mCardinal (Fig 1C), or fixed GFP-CAAX embryos at various lumen sizes ranging from 0 to $5*10^3$ $\mu m^2$ and measured the percentage of KV cells that had lumenal cilia (Fig 1E) and cilia length (Fig 1F). We found from both the live (Fig 1C) and fixed cell analysis (Fig 1E and 1F) that cilia dock at the apical membrane during early lumen formation and then extend into the lumen (Fig 1C) once lumen area approaches approximately 300 $\mu m^2$ (Fig 1E). We then compared these studies to when cilia start to elongate (Fig 1F). We find that cilia, when inside a KV cell, can reach a length of approximately 2 $\mu m$, but once a lumen is formed (300 $\mu m^2$ in area), the cilia can extend into the lumen and grow to their final approximate length of over 4 $\mu m$ (Fig 1F).

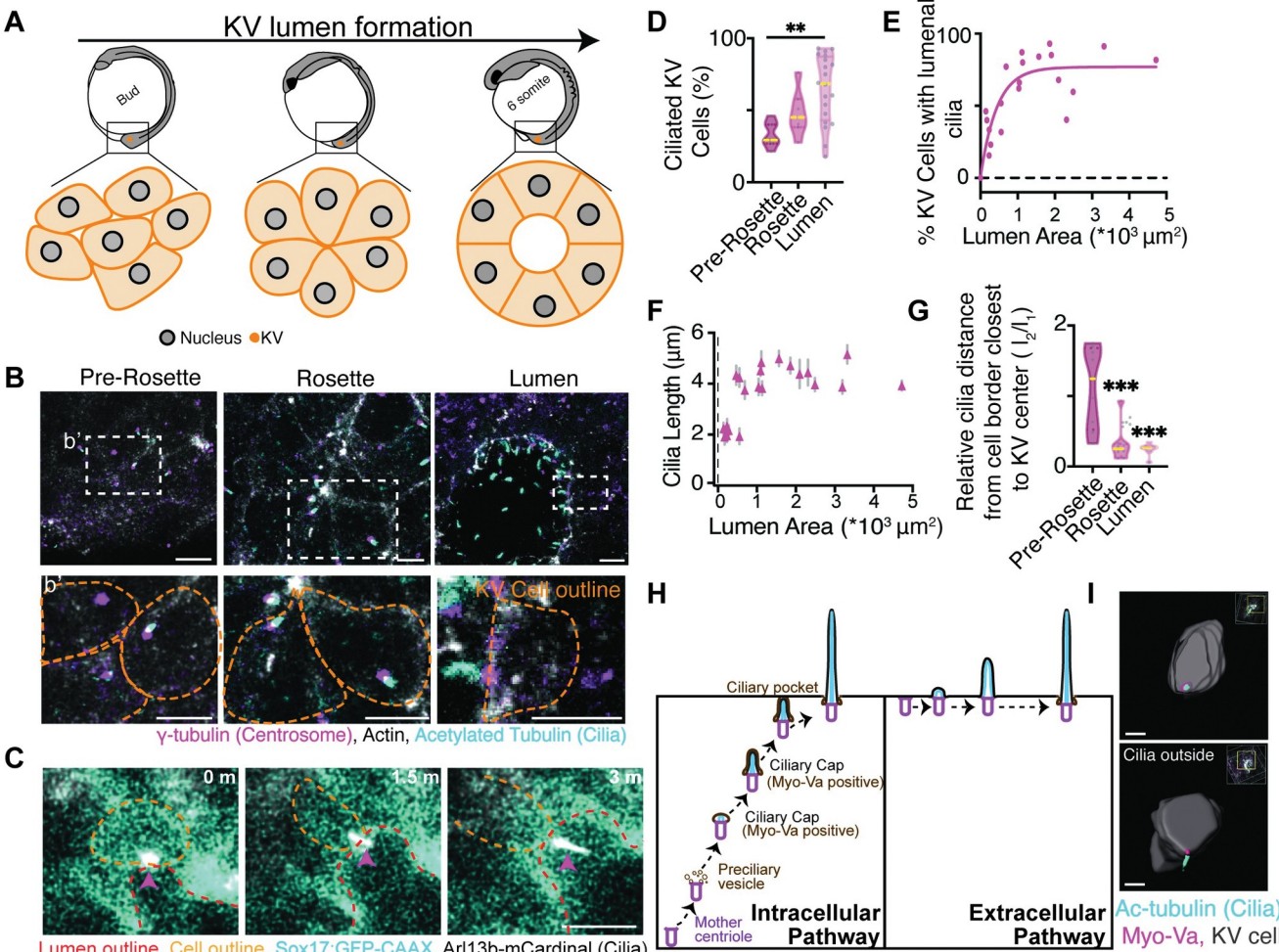

**Fig 1. KV cilia form prior to KV lumen formation using an intracellular pathway.** (**A**) Model depicting KV lumen formation across developmental stages of the zebrafish embryo. (**B**) Confocal micrographs of KV developmental stages with cilia (acetylated-tubulin, cyan), centrosome (γ-tubulin, magenta), and actin (phalloidin, gray). Scale bar, 10 μm. (**b'**) Magnified insets from (**B**) depicting centrosome and cilia positioning in KV cells at different KV developmental stages. Scale bar, 7 μm. (**C**) KV cell building and extending a cilium (Arl13b-mCardinal, gray) into the lumen of the KV. KV plasma membranes (Sox17:GFP-CAAX, cyan) shown. Scale bar, 5 μm. (**D**) Percentage of ciliated KV cells at the different KV developmental stages shown as a violin plot with median (yellow line). One way ANOVA across KV developmental stages, n>7 embryos, **p<0.01. (**E**) Scatter plot demonstrating the percentage of KV cells with lumenal cilia per embryo in relation to KV lumen area. n = 29 embryos. Goodness of fit $R^2$ = 0.8577. (**F**) Scatter plot depicting average cilia length within KV cells per embryo across n = 29 embryos in relation to lumen area. Error bars, ± SEM. (**G**) Violin plot depicting relative distance of cilia from cell border closest to KV center. median (yellow line). One way ANOVA across KV developmental stages, ***p<0.001. (**D-G**) Please refer to S1 Table for additional statistical information. (**H**) Model demonstrating intracellular versus extracellular pathways for cilia formation. (**I**) 3D surface rendering of representative KV cells with cilia (acetylated-tubulin, cyan) inside versus outside of KV plasma membranes (KV membranes, Sox17: GFP-CAAX, gray), Myo-Va (magenta). Scale bar, 5 μm.

Our findings suggest a cellular mechanism where cilia first are formed through an intracellular pathway, which can then extend into the lumen (Fig 1H). To further test that cilia were forming through an intracellular pathway, as opposed to an extracellular pathway, volumetric projections of surface rendered KV cells were performed at the pre-rosette, rosette, and lumen stages. KV cell outlines were obtained using GFP-CAAX and cilia were immunostained using acetylated tubulin (S1B Fig) along with a marker for the ciliary membrane cap, Myosin Va (Myo-Va, [29,30], Fig 1I). Surface rendering using IMARIS software allowed for the spatial positioning of cilia in KV cells across KV developmental stages to be assessed (S1B Fig and S1 Video). We identified that as KV develops from pre-rosette to rosette, then to lumen

formation, intracellular cilia surrounded by Myo-Va approach the apical membrane (Fig 1I and S1 Video). Once a lumen is formed, the cilia extend into the developing KV lumen (Figs1I and S1B and S1 Video). Once the cilium extended into the lumen, Myo-Va remained at the cilium's base (Fig 1I).

To determine if KV cell cilia were positioning towards the center of the KV cellular mass over the course of its development, we calculated the relative distance of cilia from the cell boundary closest to the KV center from the embryos shown in Fig 1B (modeled in S1C Fig; calculations in Fig 1G). When values approach 0, cilia are approaching the cell boundary closest to the KV center. This occurs significantly as KV cells transitioned from a pre-rosette organization to KV cells organized around a fluid filled lumen (Fig 1G). This suggests that KV cell cilia are constructed intracellularly, then positioned to the cell boundary closest to KV center where they are primed to extend their cilia into the forming lumen. We propose a potential mechanism by which KV cell cilia form through an intracellular pathway that recruits pre-ciliary vesicles positive for Myo-Va. These Myo-Va vesicles then form a ciliary cap for the cilia to grow within. The cilia with associated cap can then fuse with the plasma membrane and KV cilia can extend into the lumen (Fig 1H, left).

## Characterization of Rab8, Rab11, and Rab35 localization during KV lumen and cilia formation

Previous work in mammalian culture systems and preliminary morpholino studies in zebrafish KV have implicated Rab8, Rab11, and Rab35 in cilia and/or lumen establishment [18,19,21–24]. However, their cellular distribution during KV development has not been investigated, nor has it been positioned in relation to KV cilia formation. Foundational studies have demonstrated that at least one of the paralogs of Rab8, Rab11, and Rab35 is broadly expressed throughout zebrafish development, including the KV [22,31–33]. In consideration of these findings, we assessed Rab8, Rab11, and Rab35 distribution in the zebrafish KV marked by the plasma membrane marker GFP-CAAX by expressing fluorescently tagged mRNA via injection (Figs 2 and 3 and S2 Video) or in an endogenously GFP tagged transgenic line of Rab11 (Figs 2D, 3A, and 3B and S2 Video, [34]). Different stages of KV development, ranging from rosette stage (1 SS, 10 hours post fertilization, hpf), to lumen stage (6 SS, 12 hpf; Fig 2A) were monitored using live (Fig 2) or fixed (Fig 3) embryo imaging preparations. With ectopic expression of mRuby-Rab8 and mCherry-Rab11 in Sox17:GFP-CAAX embryos we identified that Rab8 and Rab11 were broadly recruited to the apical membrane during rosette formation and remained there during lumen opening (Fig 2B and 2C and S2 Video). This is consistent when compared to an endogenously tagged GFP-Rab11 line shown in Fig 2D. mRuby-Rab35 had a different distribution pattern to Rab11 and Rab8, where it primarily localized to cell-cell boundaries with some intracellular distribution (Fig 2D).

To examine localization of Rab8, Rab11, and Rab35 with cilia, GFP-Rab11 transgenic embryos that ectopically expressed mRuby-Rab8 were fixed at the KV rosette stage (Fig 3A, top) and lumen stage (Fig 3A, bottom, and 3B and 3C), and cilia were immunostained for acetylated tubulin. At the KV rosette stage cilia are organized intracellularly surrounded by Rab11 membranes organized at the center of the rosette, while Rab8 is organized at the base of the cilia where the centrosome resides (Fig 3A). As the KV develops to a lumen stage, Rab11 reorganizes to the base with Rab8 (Fig 3A and 3B). Throughout KV developmental stages, Rab35 organized to cell boundaries with no specific localization to the cilia itself (Fig 3C). These findings suggest that Rab11 and Rab8 associate with the cilia during its formation, and Rab8, Rab11, and Rab35 all are located at the forming apical membrane during lumen formation.

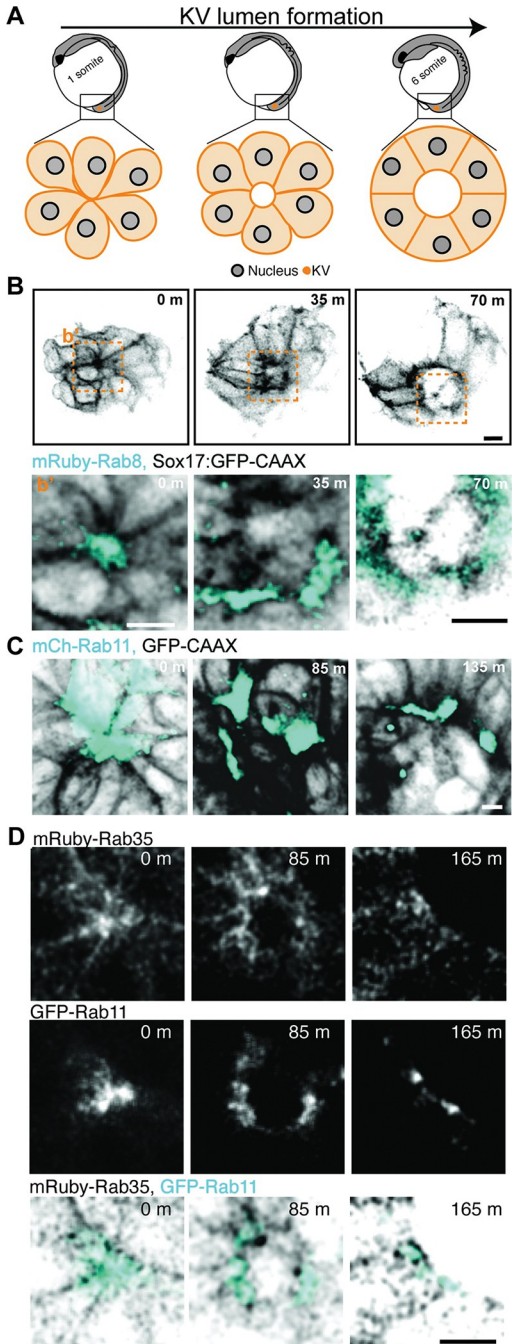

**Fig 2. Characterization of Rab8, Rab11 and Rab35 localization during KV lumen formation.** (**A**) Model depicting rosette, early lumen and late lumen stages of KV development being examined in live embryos represented in panels (**B-D**). (**B-D**) Representative images from live confocal videos of ectopically expressed mRuby-Rab8 (cyan, **B**), mCherry-Rab11 (cyan, **C**), endogenously tagged GFP-Rab11 (gray, cyan in merge, **D**) and ectopically expressed mRuby-Rab35 (gray, inverted gray in merge, **D**) localization in KV cells marked by GFP-CAAX (inverted gray, **B**) during KV lumen formation. Scale bar, 10μm. Refer to S2 Video.

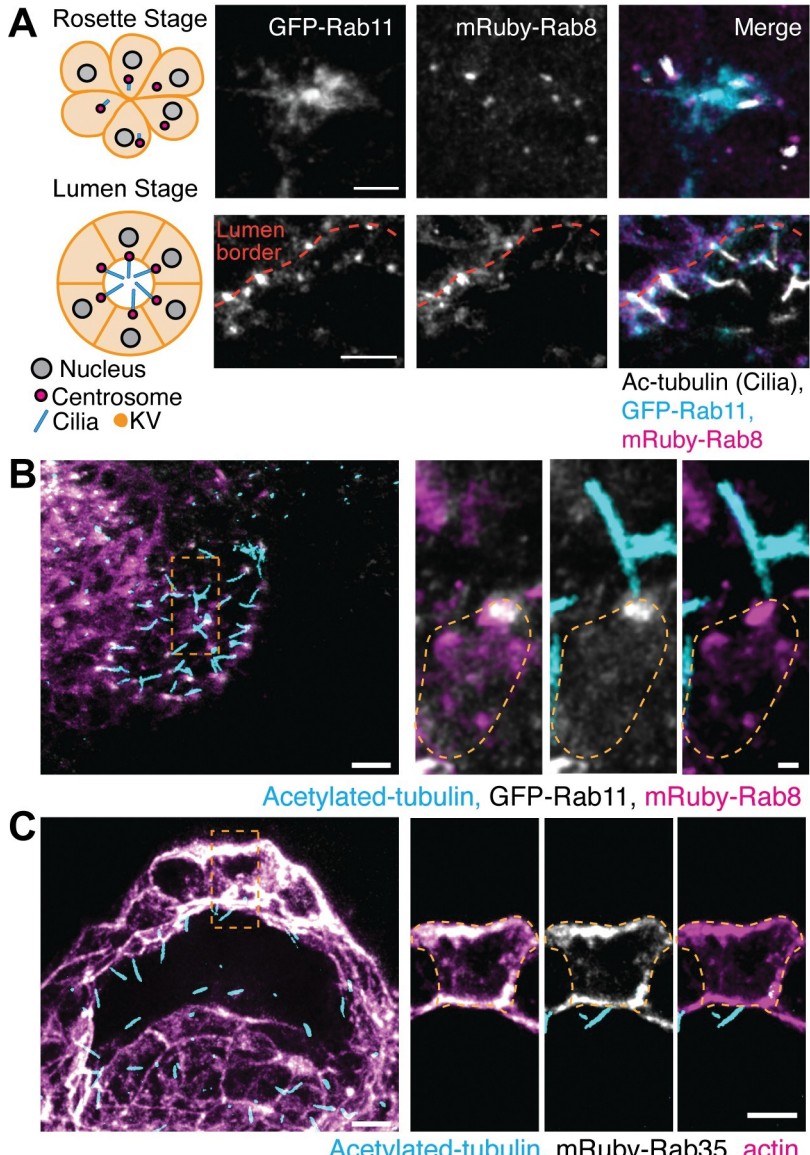

**Fig 3. Characterization of Rab8, Rab11 and Rab35 localization during KV cilia formation.** *(A)* Left, model of KV developmental stages, rosette (top) and lumen (bottom), with centrosome (magenta) and cilia (cyan) positioning. Right, confocal micrographs with GFP-Rab11 (gray, cyan in merge), mRuby-Rab8 (gray, magenta in merge), and cilia (acetylated-tubulin, gray), shown. Scale bar, 10 μm. **(B-C)** Confocal micrographs of KV lumen stage with cilia (acetylated-tubulin, cyan), GFP-Rab11 (gray, **B**), mRuby-Rab8 (magenta, **B**), mRuby-Rab35 (gray, **C**), and actin (magenta, **C**). Scale bar, 2 μm.

## Cilia extension into the KV lumen requires Rab11 and Rab35 associated membranes, but not Rab8

To test the requirement of Rab11, Rab8, and Rab35 for KV cell cilia formation we employed two strategies, acute Rab GTPase optogenetic clustering (modeled in Fig 4A) and morpholino (MO) transcript depletion using MOs that have been previously characterized ([18,22,35], S2A Fig). Since Rab11, Rab8, and Rab35 are broadly expressed during zebrafish embryo development [22,31–33], we chose to employ an optogenetic strategy to acutely inhibit their function

during KV development. This optogenetic strategy causes an acute inhibition of CIB1-Rab11-, CIB1-Rab8-, and CIB1-Rab35-associated membranes through a hetero-interaction between cryptochrome 2 (CRY2) and CIB1 upon exposure to blue light during KV developmental stages [36–38]. Previous studies identified that upon blue light exposure, CIB1-Rab5 or CIB1-Rab11-associated membrane compartments cluster together creating an intracellular traffic jam and inhibiting the specific Rab's membrane compartment from sorting intracellular cargo and regulating cellular functions [36–38]. Our studies herein find that optogenetically clustering Rab11- or Rab35-membranes during early KV development caused significant defects at 6 SS when many of the cells should be ciliated. Under control conditions (CRY2 injected) and Rab8 clustered conditions, 78.03±3.86% and 64.43±3.85% of KV cells formed cilia respectively, whereas Rab11 and Rab35 clustered embryos had a significant decrease in the percentage of ciliated cells (35.81±8.79% for Rab11, 49.72±5.50% for Rab35, Figs 4B, 4C, and S2B). KV cells that could form cilia under Rab11- and Rab35-clustered conditions had

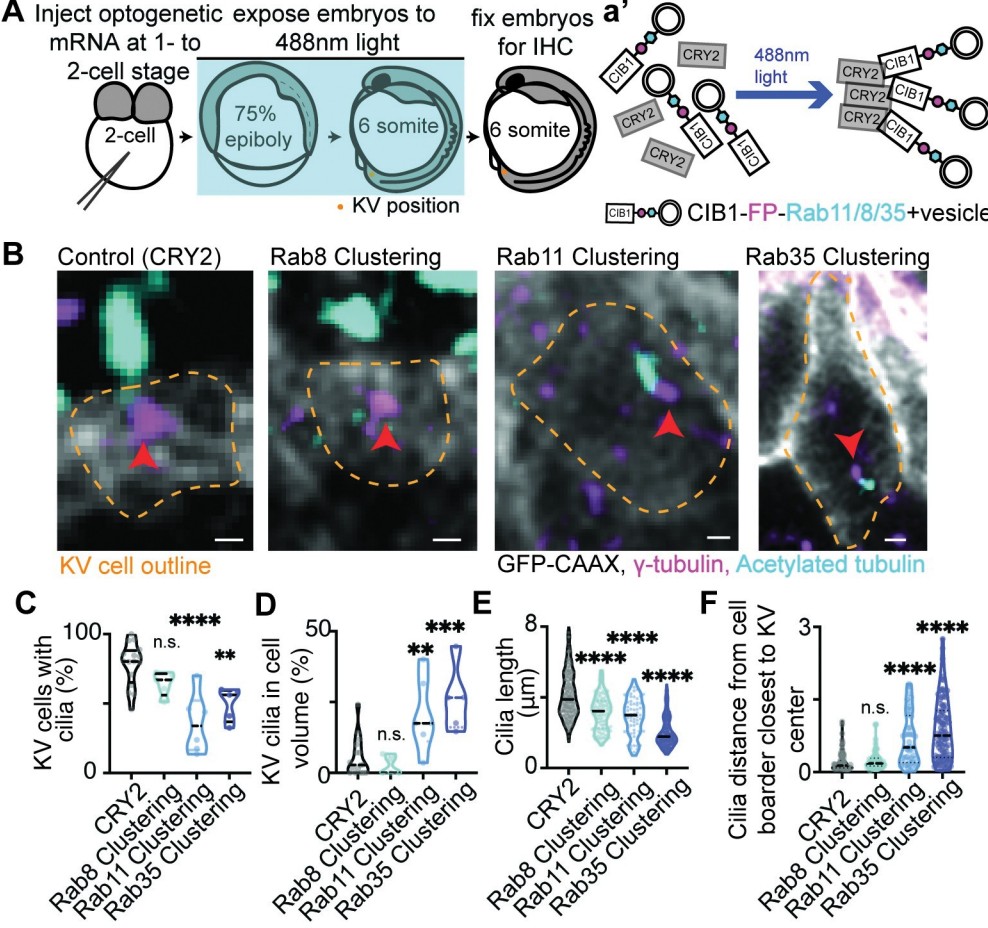

**Fig 4. Cilia extension into the KV lumen requires Rab11- and Rab35-associated membranes, but not Rab8.** (**A**) A model depicting the use of optogenetics to acutely block Rab GTPase-associated trafficking events during KV developmental stages. (**B**) Confocal micrographs of cilia (acetylated tubulin, cyan) in CRY2 (control), Rab8-, Rab11-, and Rab35-clustered Sox17:GFP-CAAX embryos (gray). Centrosomes denoted by γ-tubulin (magenta). Clusters not shown. Yellow dashed lines are KV cell cortical membranes. Orange arrow, centrosome. Scale bar, 2μm. (**C-F**) Violin plots of percentage of KV cells with cilia (**C**), percentage of KV cilia in cell volume (**D**), cilia length (**E**), and the relative distance of cilia from the cell boarder closest to KV center (**F**). One way ANOVA with Dunnett's multiple comparison to CRY2 (control) was performed. n>4 embryos, n.s. not significant, **p<0.01, ***p<0.001, ****p<0.0001. Statistical results detailed in S1 Table.

most of their cilia stuck in the cell volume (Fig 4B and 4D). Rab11-, Rab8-, and Rab35-clustered cells that made cilia demonstrated significantly decreased cilia length (2.92±0.14 μm for Rab11, 3.15±0.08 μm for Rab8, and 2.02±0.05 for Rab35) compared to control CRY2 conditions (4.13±0.06 μm, Fig 4E). This significant decrease in cilia length with Rab11, Rab8, and Rab35 clustering, is consistent with Rab11, Rab8, or Rab35 depletion using morpholinos (S2A and S2C Fig). Interestingly, even though Rab8 clustering caused a significant decrease in cilia length (Fig 4E), it was dispensable for the formation of cilia or its extension into the KV lumen (Figs 4C, 4D, and S2B). These findings suggest that centrosomes that construct a cilium under Rab11- and Rab35-clustering conditions are unable to extend the cilium into the lumen and that this could be the underlying reason that cilia are shorter in length. To test this idea, cilia length was measured in cells that had cilia still in the cell volume under clustering conditions. Significant length defects were observed under Rab11, Rab35, and Rab8 clustering conditions (S2D Fig) when compared to the length of formed cilia that remained in the cell volume at early KV development stages measured in Fig 1F (shown as dashed line in S2D Fig). When comparing cilia stuck in the cell volume to cilia that are lumenal, we found that the cilia in the cell volume under clustering conditions were significantly shorter in length than ones that could extend into the lumen (S2D Fig). We argue that cilia can only extend to a certain length when inside the cell, and then extends to its final length when positioning into the lumen. Our studies suggest that Rab11, Rab8, and Rab35 are all involved in general building of the cilium. However, Rab8 is dispensable for cilia positioning into the forming KV lumenal space, whereas Rab11 and Rab35 are necessary.

One potential reason for cilia not extending into the lumen is that the centrosomes with associated cilia are not able to position at a forming apical membrane. To test the role of Rab35-, Rab11- and Rab8-membranes in intracellular cilia positioning during KV development, the associated centrosome distances from the plasma membrane closest to KV center were measured under clustered conditions and compared to control conditions (CRY2). If centrosomes are positioning towards the KV center, then the number should approach 0. Rab11- and Rab35-clustered embryos measurements averaged around 0.70±0.10 and 0.75 ±0.11 respectively, whereas with Rab8 clustered and control conditions the centrosome distance approached 0 with a value of 0.23±0.06 and 0.30±0.03 (Fig 4F). These studies suggest that Rab11 and Rab35 coordinate centrosome positioning and cilia formation during KV development.

## Rab11 and Rab35 modulate KV lumen formation

Our initial studies demonstrated that KV cilia extend into the lumen once the lumen reaches an area 300 μm$^2$ (Fig 1E). Then KV cilia can reach their maximum length of approximately 4 μm (Fig 1F). These findings suggested that mechanisms regulating cilia formation may also play an important role in coordinating lumen formation. We tested the requirement of Rab11, Rab8, and Rab35 for KV lumen establishment using MO transcript depletion (Figs S2A, S3A, and S3B) and optogenetic clustering (Figs 4A and 5A-D). With acute optogenetic clustering of Rab11- and Rab35-associated membranes in Sox17:GFP-CAAX embryos (Fig 5A) we identified severe defects in KV lumen development that was consistent when depleting transcripts using MOs (S3A and S3B Fig) when comparing to control conditions (CRY2, Fig 5A; control MO, S3A and S3B). This was measured both by following lumen formation live using an automated fluorescent stereoscope set up for a set time frame (Fig 5A and S3 Video) and at a fixed developmental endpoint (6 SS, Fig 5C and 5D). For live embryo analysis, Sox17:GFP-CAAX embryos were imaged just past 75% epiboly for over 4 hours, during this time, the Rab35 and Rab11 clustered embryos were not able to form a lumen when compared to control (CRY2) or

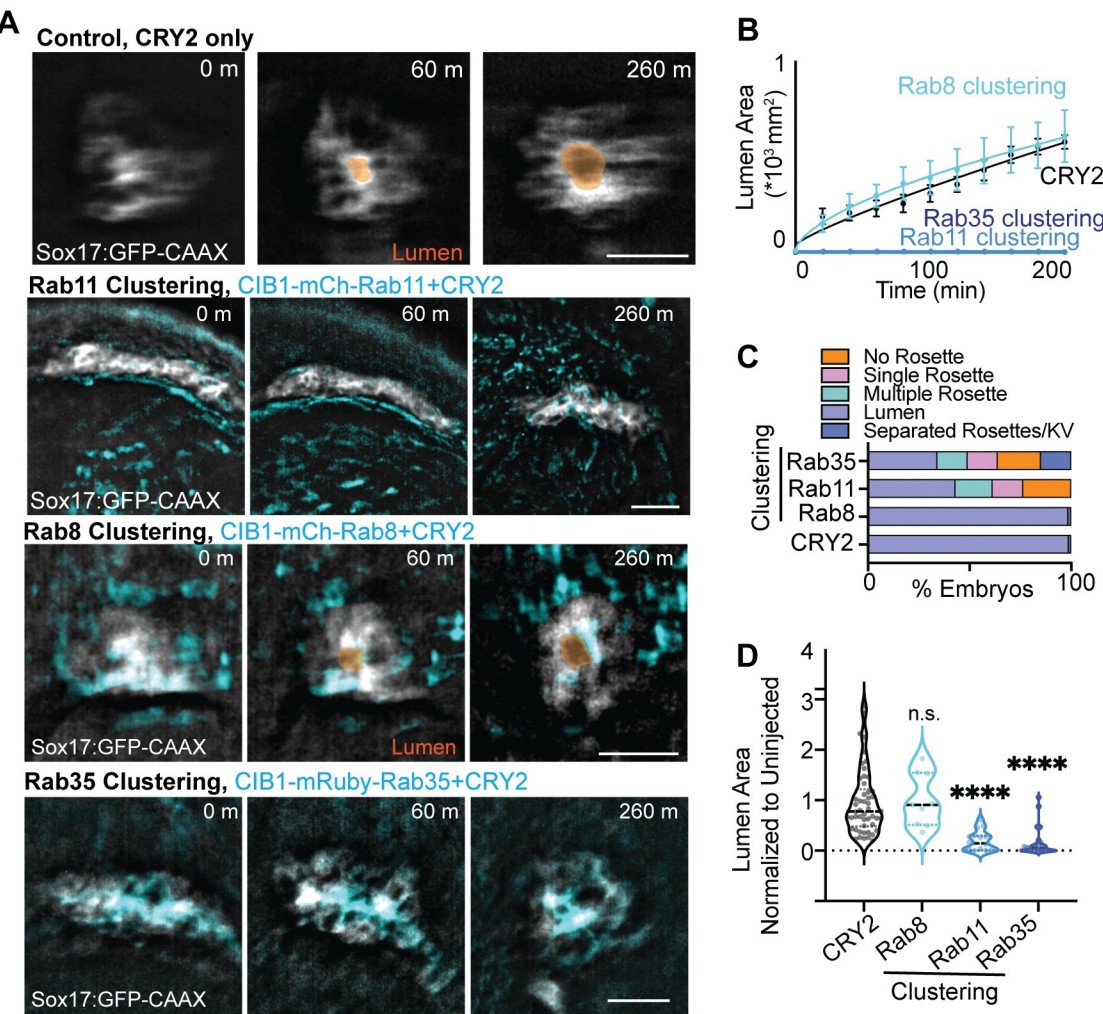

**Fig 5. Rab11 and Rab35 modulate KV lumen formation. (A)** Optogenetic clustering of Rab11 and Rab35 blocks KV lumen formation compared to CRY2 control and Rab8. Imaged on an automated fluorescent stereoscope. Scale bar, 50 μm. KV marked with Sox17:GFP-CAAX, lumens highlighted in orange, clusters shown in cyan. Refer to S3 Video. **(B)** KV lumen area over time (±SEM for n = 3 embryos per condition) in control (CRY2 injection) and Rab8, Rab11 and Rab35 clustering conditions. **(C)** KV morphologies characterized from optogenetically-clustered then fixed embryos at 12 SS (12 hpf). n>47 embryos per condition measured across n>9 clutches. **(D)** Violin plot depicting lumen area from Rab8, Rab11, and Rab35 clustering conditions normalized to uninjected control values. Dots represent individual KV values. Median denoted by line. One-way ANOVA with Dunnett's multiple comparison test, compared to CRY2. n>9 embryos, n.s. not significant, ****p<0.0001. Statistical results detailed in S1 Table.

Rab8-clustered embryos (Fig 5B and S3 Video). When assessing at a fixed developmental end-point (6 SS, 12 hpf), we found that Rab11 and Rab35 clustered embryos presented with defects in forming a rosette (23.7% of embryos for Rab11, 21.3% for Rab35) or transitioning from a multiple rosette state to a single rosette state (18.3% of embryos for Rab11, 14.9% for Rab35, Fig 5C) compared to CRY2 embryos or Rab8 clustered embryos (98.8% and 98.6% form lumen, Fig 5C and S3 Video). With Rab35 and Rab11 clustering, less than 50% of KVs were able to form a lumen (Fig 5C), and the lumens they did form were significantly decreased in size (Figs 5D and S3B). While a Rab11-Rab8 GTPase cascade during lumen formation has been proposed in the context of mammalian cell culture conditions [21], our findings were surprising in that acute Rab8 clustering conditions or Rab8 depletion conditions by MO did

not affect lumen formation during KV development. Instead, Rab11 and Rab35 played a predominant role in this process.

### Rab11 and Rab35, but not Rab8, mediates CFTR trafficking to the apical membrane

Since optogenetic clustering of both Rab11 and Rab35, but not Rab8, resulted in lumen formation defects, we wanted to examine whether they disrupted CFTR recruitment to the apical membrane. CFTR is a master regulator of fluid secretion into lumenal spaces. CFTR is transported through the secretory pathway to the apical membrane where it mediates chloride ion transport from inside the cell to the outside. Loss of CFTR-mediated fluid secretion impairs KV lumen expansion leading to laterality defects [12]. Our studies find that Rab11 optogenetic clustering causes a severe defect in CFTR delivery to the apical membrane where CFTR-GFP becomes trapped in Rab11- and Rab35-clustered membrane compartments (Fig 6A and 6B). With both Rab11 and Rab35 clustering, there was significantly less CFTR that was able to be delivered to forming apical membranes. This is consistent with defects in KV rosette and lumen formation observed with Rab11 and Rab35 clustered embryos (Fig 5C and 5D).

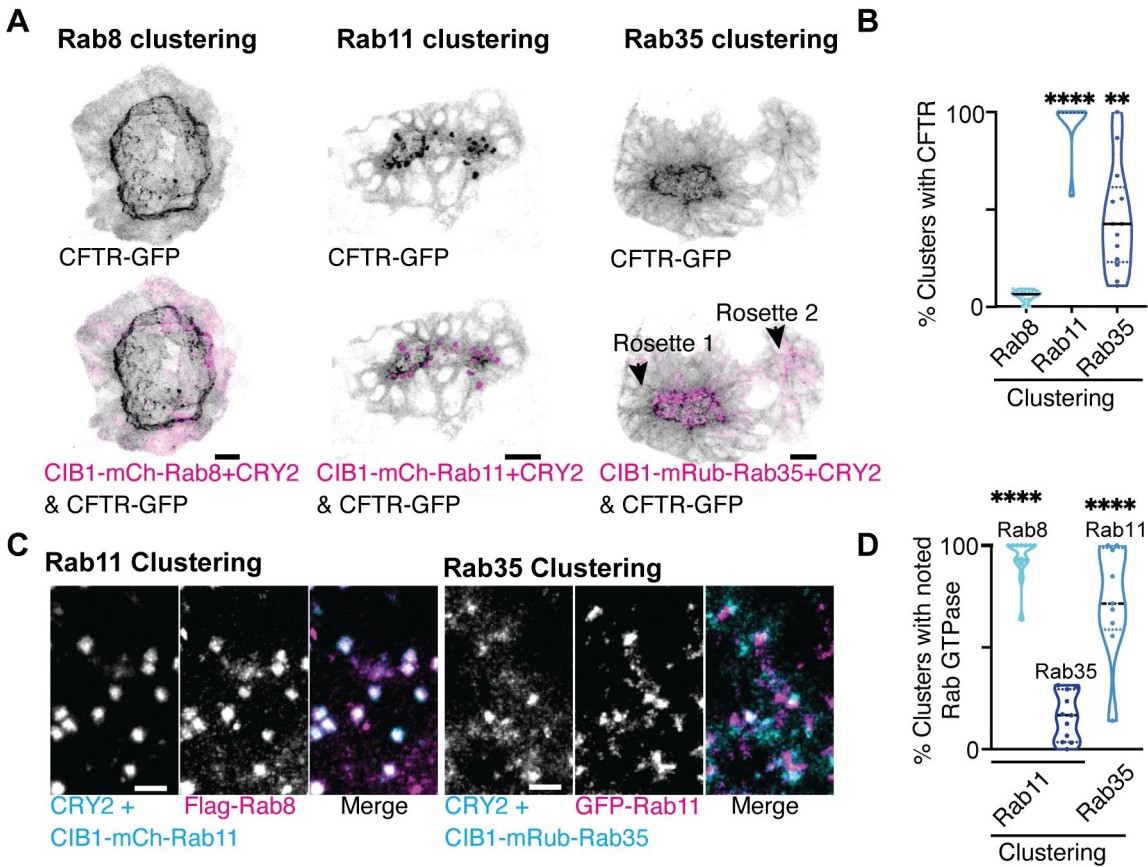

**Fig 6. Rab11 and Rab35, but not Rab8, regulates KV lumen formation by mediating CFTR trafficking to the apical membrane.** (**A**) Optogenetic clustering of Rab11, Rab8, and Rab35 (magenta) in KV cells. Localization with CFTR-GFP (inverted gray) is shown. Scale bar, 20 μm. (**B**) Violin plots depicting percent of optogenetic clusters that colocalize with CFTR. n>9 embryos, **p<0.01, ****p<0.0001. (**C**) Optogenetic clustering of CIB1-mCherry-Rab11 and CIB1-mRuby-Rab35 (cyan). CIB1-mCherry-Rab11 clusters localization with Flag-Rab8 (magenta) or CIB1-mRuby-Rab35 clusters with GFP-Rab11 (magenta) shown. Scale bar, 7 μm. (**D**) Violin plot depicting percent of optogenetic clusters that colocalize with Rab8, Rab35, or Rab11. n>9 embryos, ****p<0.0001. Statistical results detailed in S1 Table.

Interestingly, some Rab35 clustered embryos assemble multiple rosettes in a KV, with one rosette being competent for lumen formation but defective in expansion (Figs 6A and S4A). When this occurs, we find that the rosette that is competent in lumen opening has some CFTR localized to the apical membrane (Rosette 1, Fig 6A), as opposed to the secondary rosette that cannot open (Rosette 2, Fig 6A). No defect in CFTR delivery to the apical membrane was noted with Rab8 optogenetic clustering (Fig 6A and 6B), consistent with the lack of observed defects in lumen size with both optogenetic clustering (Fig 5) and depletion of Rab8 using morpholinos (S3A and S3B Fig).

Some GTPases are known to work together on the same membrane compartment. For instance, Rab11 and Rab8 were reported to function together in a GTPase cascade on recycling endosomes to regulate cellular events such as lumen formation and ciliogenesis. In this situation, Rab11 acts upstream of Rab8 by recruiting the Guanine Exchange Factor (GEF) for Rab8, Rabin8 [18,19,21]. Based on our findings that both Rab11 and Rab35 cause defects in lumen formation and CFTR trafficking, we asked if Rab11, Rab35, and/or Rab8 could act on the same membrane compartment. To test this, we performed optogenetic clustering of Rab35 or Rab11 and determined whether clustering one recruited Rab11, Rab35, or Rab8 to the cluster. Optogenetic clustering of Rab11 resulted in the recruitment of Rab8 but not Rab35 (Figs 6C, 6D, and S4B). This is consistent with the idea that a Rab11 cascade may still exist between Rab11 and Rab8, but that this cascade is not required for CFTR transport or KV lumen formation. It also suggests that Rab11 is not acting upstream of Rab35. Interestingly, upon optogenetic clustering of Rab35 membranes, Rab11 becomes co-localized (Fig 6C and 6D) suggesting that Rab35 is upstream of Rab11. In summary, we find that Rab35 may act upstream of Rab11 to ensure appropriate lumen formation and subsequent expansion through managing CFTR trafficking to the forming apical membrane.

## Discussion

While select Rab GTPases have been extensively studied, most of them have not been assigned a detailed function or localization pattern during early embryonic vertebrate development. Because Rab GTPases are potentially required for a variety of cellular functions and developmental contexts, we needed to employ a strategy to acutely disrupt their function. Here, we used an optogenetic strategy that takes advantage of Rab GTPases membrane association, where the Rab GTPase of interest is attached to CIB1, and we express CIB1's optogenetic binding partner CRY2. Upon exposure to blue light, CIB1 forms heteromeric complexes with CRY2 causing the Rab associated membranes to cluster together and become non-functional (modeled in Fig 4A). This approach is versatile in developmental models due to its acute triggering and reversibility. For instance, we can acutely cluster Rab-associated membranes during a specific developmental stage, and then release the clustering through the removal of blue-light and examine downstream developmental consequences. Zebrafish embryos are an ideal developmental system for this work due to their optical transparency and external development making them easily accessible to blue light exposure [37,38]. In these studies we focused on three Rab GTPases (Rab8, Rab11, and Rab35) that have been linked to lumen and cilia formation in mammalian cell culture models [18,19,21,24]. We used a combination of optogenetic and traditional depletion approaches (using MO) to examine the roles of these GTPases *in vivo* during left-right organizer development.

Previous foundational studies identified that cells within the left-right organizer (KV) assemble into a cyst like structure surrounding a fluid filled lumen, with the majority of KV cells having motile cilia [9,17,39]. Motile cilia in the KV lumen direct fluid flow, which is essential for the establishment of the embryo's left-right axis [5]. However, when KV cells start

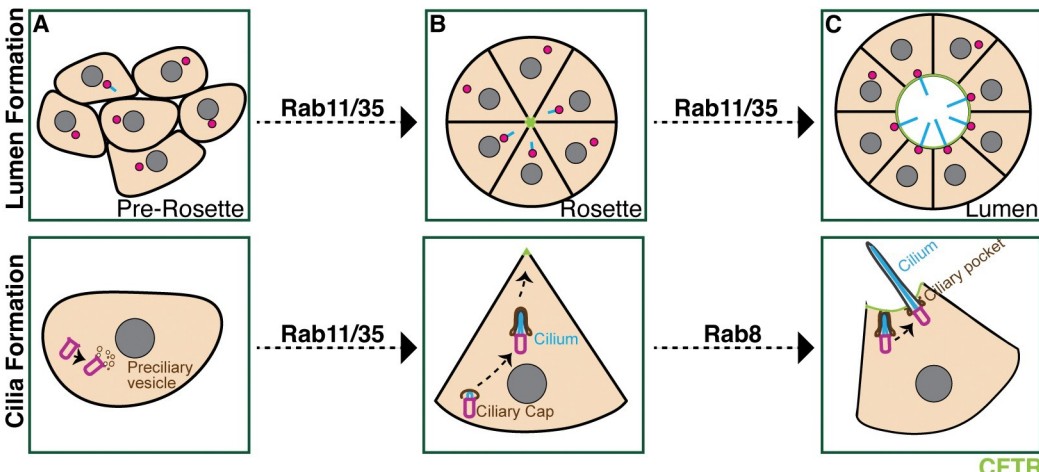

**Fig 7. Model depicting KV lumen and cilia formation across KV developmental stages.** Centrosome depicted in magenta, cilia in cyan and CFTR in green. (**A-B**) At the pre-rosette stage (**A**) a proportion of centrosomes start to assemble cilia that then reposition towards the center of the KV at the rosette stage (**B**) in a Rab11 and Rab35 dependent manner. In (**B**), Rab11 and Rab35 mediate CFTR transport to the apical membrane. (**C**) The rosette stage then transitions to a lumen stage where most of the centrosomes locate at the CFTR-positive apical membrane and extend their cilia into the lumen where cilia elongate to their full length in a Rab8 dependent manner.

to form cilia and how cilia formation is coordinated with KV lumen formation have yet to be ascertained. Our studies establish that KV precursor cells, DFCs, assemble a cilium inside the cell before the KV cells start to assemble into a cyst-like structure (Fig 7A and 7B). We noted that cilia and their associated centrosomes reposition inside the cell towards the center of the KV cellular mass while KV cells are rearranging into a rosette like structure (Fig 7A and 7B), a pre-requisite structure that precedes lumen formation. This movement of the centrosome and rearrangement into a rosette like structure relies on the small GTPases Rab11 and Rab35 (Fig 7). Specifically, we find that Rab35 acts upstream of Rab11 on the same membrane compartment, likely recycling endosomes [40], to assist in expanding the lumen by regulating the delivery of CFTR to the apical membrane (Figs 6 and 7C). Previous foundational work identified that CFTR recruitment to the apical membrane is a requirement for lumen expansion [12]. We show that once the lumen expands to a specific area (Fig 1), which is mediated by Rab11 and Rab35 (Fig 7), the KV cilia can extend and elongate into the lumen in a Rab8 dependent manner (Fig 4).

Interestingly, the only significant defect we found with acute disruption of Rab8 was cilia length (Fig 4E), whereas with Rab11 and Rab35 we found defects in KV development that included rosette formation and lumen formation (Fig 5), along with a defect in cilia formation (Fig 4). This was surprising due to previous reports identifying a GTPase cascade between Rab11 and Rab8 that was needed for lumen formation and cilia formation in mammalian tissue culture [18,19,21,27,28]. While we argue that this cascade may not be required for lumen or cilia formation in KV cells, it may still be intact and is involved in regulation of cilia length (Figs 4 and 7). Our findings demonstrate that both conserved and divergent mechanisms are likely involved in cilia formation dependent on the developmental requirements of the tissue being formed. For instance, there may be a possible connection between Rab35 and Rab11 that is coordinated during cilia and lumen formation, where both Rab35 and Rab11 clustered membranes result in the sequestration of CFTR (Fig 6A and 6B), and that Rab35 clustering results in the partial recruitment of Rab11 (Fig 6C and 6D). Interestingly, there is no colocalization between Rab35 and cilia (Fig 3), unlike Rab11 (Fig 3A).

One potential unique mechanistic possibility is that Rab35 and Rab11 work together in coordinating lumen formation through CFTR transport (Fig 6). In support of this scenario, Rab11 or Rab35 clustering prevents CFTR from accumulating appropriately at the apical membrane, resulting in incomplete lumen formation that may consequently cause cilia to remain inside the cell. Our studies find that with acute inhibition of Rab11 and Rab35 associated membrane compartments, a significant increase in KV cells have internalized cilia compared to control and Rab8 membrane inhibition (Fig 4D). These same conditions cause defects in lumen formation (Fig 5A and 5B) and the KVs that do form a lumen are significantly decreased in area (Figs 5D and S3B). However, we do not think that CFTR transport to the apical membrane is solely required for lumen formation due to key studies that demonstrate that inhibition of CFTR does not block lumen formation, but disrupts lumen expansion [10,12]. One specific study demonstrated that CFTR mutations (*pd1048*) and treatment with morpholinos against CFTR did not affect cilia formation, but did result in lumen area defects at 8 SS [10]. On average lumen areas were approximately 1000 $\mu m^2$ in *cftr$^{pd1048}$* mutant fish compared to 3000 $\mu m^2$ in controls at 8 SS. This suggests that a moderate decrease in lumen area caused by loss of CFTR function does not directly affect cilia formation. On this topic, we do not find lumen expansion defects with Rab8 clustering or depletion (MO), but we do find cilia length defects that also suggests that cilia length is not solely dependent on lumen area dynamics. For future studies, we are considering investigating if early Rab11 and Rab35 mediated trafficking events to the centrosome during cilia construction (Fig 7A) are required for lumen formation, or perhaps Rab11 and Rab35 have two independent roles one in cilia formation and one in lumen formation.

An additional conserved mechanism for Rab11, like what is reported in mammalian cell culture, is a direct role at the cilium where Rab11 localizes to (Fig 3A). In this scenario, Rab11 regulates cilia formation and elongation in a cascade with Rab8. We argue that this cascade is likely in place based on our findings that acute inhibition of Rab11-associated membranes through optogenetic clustering recruits Rab8 to these membranes, but clustering Rab8 does not recruit Rab11. These findings suggest that Rab11 is upstream of Rab8 and can recruit Rab8 to the same membrane compartment to potentially regulate cilia elongation (Fig 6C, modeled in Fig 7).

Our findings demonstrate that both conserved and divergent mechanisms for cilia formation likely exist, and Rab GTPases relative roles are likely dependent on the developmental requirements of the tissue being formed. Our studies validate zebrafish to be a versatile model to identify the potential mechanisms of function for Rab GTPases *in vivo*.

## Experimental procedures

### Experimental model and subject details

**Fish lines.**   Zebrafish lines were maintained using standard procedures approved by Syracuse University IACUC (Institutional Animal Care Committee) (Protocol #18–006). Embryos were raised at 28.5˚C and staged (as described in [41]). Wildtype and/or transgenic zebrafish lines used for live imaging and immunohistochemistry are listed in key resource table (S2 Table).

## Method details

### Ethical statement

Animal studies were performed with respect to the guidelines and standard procedures detailed by the Institutional Animal Care and Use Committee (IACUC) at Syracuse

University. All protocols involving live vertebrate animals were approved by Syracuse University IACUC (Protocol #18–006). Rapid chilling or tricaine was used to euthanize Zebrafish following IACUC guidelines.

## Antibodies

Antibody catalog information used in mammalian cell culture and zebrafish embryos are detailed in key resource table (S2 Table).

## Plasmids and mRNA

Plasmids were generated using Gibson cloning methods (NEBuilder HiFi DNA assembly Cloning Kit) and maxi-prepped before injection and/or transfection. mRNA was made using mMESSAGE mMACHINE SP6 transcription kit. See key resource table (S2 Table) for a list of plasmid constructs and mRNA used.

## Morpholinos

Morpholinos (MO) were ordered from Gene Tools. Previously characterized Rab8, Rab11, and Rab35 MO sequences were used from [18,22,28]. Control morpholinos were prepared vivo standard control morpholinos from Gene tools. This negative control oligo targets an intron mutation of human beta-globin. See Supplementary key resource table in S2 Table for a list of morpholinos used.

## RNA extraction and RT-PCR

Total RNA was extracted from either an isolated embryo or several embryos injected with control, Rab8, Rab11 or Rab35 morpholinos using TRIzol reagent. The RT-PCR was performed on each sample using OneTaq One-Step RT-PCR Kit (see key resource table, S2 Table) with the forward primers "tcagtatggcgaagacctacgat", "gttagcatggctactgcctaatcac", "gtaatgagcgact-gactgctgac" and reverse primers "tcttcacagtagcacacagcga", "catgtcattgtctcggcggtc", "gtgcaagga-gaaaaataagatcaagttagagaatca" for Rab8, Rab11 and Rab35 consecutively. RT-PCR reaction was run using the following cycling conditions: 48˚C for 30 min, 94˚C for 1min followed by 40 cycles of 94˚C for 15 sec, 54˚C (Rab8 and Rab11) or 53˚C (Rab35) for 30 sec, 68˚C for 2 minutes with final extension at 68˚C for 5 min.

## Immunofluorescence

Fluorescent transgenic and/or mRNA injected embryos (refer to strains and mRNAs in key resource table (S2 Table), and for injection protocols refer to [42,43]) were staged at Kupffer's Vesicle (KV) developmental stages as described in [37,44] and fixed using 4% paraformaldehyde with 0.1% triton-100. Standard immunofluorescent protocols were carried out (refer to [43,45]). Embryos were then embedded in low-melting 2% agarose (see key resource table, S2 Table) with the KV positioned at the bottom of a #1.5 glass bottom MatTek plate (see key resource table, S2 Table) and imaged using a laser scanning confocal microscope (see details below). Live KV lumen formation videos were accomplished by using agar molds and orienting the KV lumen towards the optics of a Leica M165 FC stereomicroscope.

## Imaging

Zebrafish embryos were imaged using Leica SP8 laser scanner confocal microscope (LSCM). The SP8 laser scanning confocal microscope is equipped with HC PL APO 20x/0.75 IMM CORR CS2 objective, HC PL APO 40x/1.10 W CORR CS2 0.65 water objective and HC PL

APO x63/1.3 Glyc CORR CS2 glycerol objective. LAS-X software was used to acquire images. A Leica M165 FC stereomicroscope equipped with DFC 9000 GT sCMOS camera was used for staging, phenotypic analysis, and live-KV lumen formation videos in zebrafish embryos. Images were acquired ever 1–10 minutes (as denoted in legends) for lumen formation experiments across the volume of the KV using either the LSCM or stereomicroscope.

### Optogenetic experiments in zebrafish embryos

Tg(sox17:GFP-CAAX), Tg*BAC*(cftr-GFP), Tg(sox17:GFP), Tg(sox17:DsRed) and TgKIeGF-P-Rab11a zebrafish embryos were injected with 50–100 pg of CRY2 and/or CIB1-mCherry-Rab11, CIB1-mCherry-Rab8 or CIB1-mRuby-Rab35 at the one cell to 4 cell stage. Embryos were allowed to develop in the dark until uninjected embryos reached the 75% epiboly stage where we can screen embryos for KV cells and expose them to 488nm light using the NIGHT-SEA fluorescence system until the six-somite stage [37,38]. Embryos were then fixed and immunostained (refer to [43,45]).

### Image and data analysis

Images were processed using FIJI/ImageJ. Graphs and statistical analysis were produced using Prism 9 software. Surface rendering (refer to [37]) and analysis of KV cells were performed using Bitplane IMARIS software. Videos were created using FIJI/ImageJ or IMARIS. Cilia length was measured as the distance from the base of the cilia to the tip using line function in IMARIS. For percentage of ciliated KV cells, the number of cells with cilia was counted and represented as a percentage over the total number of cells in the cyst forming tissue.

*Relative cilia distance from cell border closest to KV center*: the distance from cilia to the cell membrane closest to KV center ($l2$) was measured and divided by the distance of the center of the cell (nucleus) to the cell's membrane closes to KV center($l1$); $d = l2/l1$ (refer to S1F Fig). This was done for KV cells with positive cilia staining at each developmental KV stage.

*Calculating colocalization of CFTR and Rab GTPases with select optogenetic clusters*: From fixed embryos the total number of Rab GTPase clusters were counted for each KV. The number of Rab clusters that had CFTR or Rab GTPase being tested overlapping with the Rab GTPase cluster was counted and presented as a percentage.

### Statistical analysis

Unpaired two-tailed t-tests and one way ANOVA were performed using PRISM9 software. **** denotes a p-value<0.0001, *** p-value<0.001, **p-value<0.01, *p-value<0.05, n.s. not significant. For further information on detailed statistical analysis see S1 Table.

### Supporting information

**S1 Video. Cilia cellular positioning in 3D.** 3D surface rendering from S1B Fig of a single KV cell at the KV pre-rosette, rosette, or lumen stage rotated 360 degrees around the X-axis. Scale bar, 5 μm. Inset shows full KV with cilia (cyan) and KV plasma membrane (Sox17: GFP-CAAX, gray). Refer to S1B Fig.
(AVI)

**S2 Video. Videos of Rab8, Rab11, and Rab35 distribution during KV lumen formation.** Live confocal videos of mRuby-Rab8 (cyan), GFP-Rab11 (gray), and mRuby-Rab35 (gray) localization in KV cells during lumen formation. KV plasma membranes (Sox17:GFP-CAAX) shown with Rab8 (inverted gray). Scale bar, 10 μm. Refer to Fig 2B–2D.
(AVI)

**S3 Video. Rab11 and Rab35 modulate KV lumen formation.** Optogenetic clustering of Rab11 and Rab35 blocks KV lumen formation compared to Rab8. Embryos imaged on automated fluorescent stereoscope every 10 min. Scale bar, 100 μm. KV marked with Sox17: GFP-CAAX, gray. Refer to Fig 5A.
(AVI)

**S1 Fig. KV cilia form prior to KV lumen formation using an intracellular pathway.** (**A**) Confocal micrographs showing actin (gray) representatives in Fig 1B. Scale bar, 10 μm. (**B**) 3D surface rendering of a representative KV cell from pre-rosette, rosette, and lumen KV developmental stages with cilia (acetylated-tubulin, cyan) and KV plasma membranes (KV membranes, Sox17:GFP-CAAX, gray). Refer to S1 Video. Scale bar, 5 μm. (**C**) Model depicting quantification of relative distance of the cilium from the cell border closest to KV center. Cilia, cyan. Nucleus, gray. Center of KV cells, yellow. Pink dashed line is distance of cilium from cell membrane. Black dashed line is distance of cell center to cell membrane.
(TIF)

**S2 Fig. Cilia extension into the KV lumen requires Rab11- and Rab35-associated membranes, but not Rab8.** (**A**) Agarose gel demonstrating RT-PCR of Rab8, Rab11, and Rab35 MO treatment compared to control MO conditions. Amplification of Rab8, Rab11, and Rab35 transcripts shown. NC, negative control. (**B**) Confocal micrographs of cilia (acetylated tubulin, cyan) in CRY2 (control), Rab8-, Rab11-, and Rab35-clustered Sox17:GFP-CAAX embryos (gray). Clusters not shown. Lumen outline is orange dashed lines. Scale bar, 10 μm. (**C**) Violin plot depicting cilia length from control (vivo standard control morpholinos from Gene tools), Rab8, Rab11, and Rab35 MO treatment. Dots represent individual cilia length values. Median denoted by line. One-way ANOVA with Dunnett's multiple comparison test, compared to CRY2. ****$p < 0.0001$. (**D**) Violin plot depicting length of lumenal cilia compared to cilia in cell volume from Rab8, Rab11, and Rab35 clustering conditions. Dots represent individual cilia length values. Median denoted by line. Dashed lines represent measurements from control embryos denoting average length of lumenal cilia (orange) and average length of cilia in cell volume (black). Unpaired student t-tests. ****$p < 0.0001$. Statistical results detailed in S1 Table.
(TIF)

**S3 Fig. Rab11 and Rab35 modulate KV lumen formation.** (**A**) Representative 3D rendering of KV under Rab8, Rab11, and Rab35 MO treatment. Lumen trace (orange), cell membrane (GFP-CAAX, inverted gray), and actin (magenta) shown. Scale bar, 25 μm. (**B**) Violin plot depicting lumen area normalized to uninjected control values in control (vivo standard control morpholinos from Gene tools), Rab8, Rab11 and Rab35 MO injected embryos. Dots represent individual KV values. Median denoted by line. One-way ANOVA with Dunnett's multiple comparison test, compared to control MO. n>12 embryos, n.s. not significant, ****$p < 0.0001$.
(TIF)

**S4 Fig. Rab11 and Rab35, but not Rab8, regulates KV lumen formation by mediating CFTR trafficking to the apical membrane.** (**A**) Representative image of optogenetic clustering of Rab35 (cyan) in KV cells; CFTR-GFP (inverted gray) shown. Scale bar, 25 μm. (**B**) Optogenetic clustering of Rab11 in KV cells. Rab11 clusters (cyan) localization with mRuby-Rab35 (magenta) shown. Scale bar, 7 μm.
(TIF)

**S1 Table. Detailed statistical analysis of results reported in this study.** Figures included are Figs 1D–1G, 4C–4F, S2C–S2D, 5B–5D, S3B, 6B and 6D.
(DOCX)

**S2 Table. Supplementary key resource table of materials used in this study.**
(DOCX)

**S1 Data. Data that underline the study.**
(ZIP)

## Acknowledgments

We thank the Michel Bagnat and Daniel Levic at Duke University School of Medicine for sharing their eGFP-Rab11 transgenic zebrafish lines.

## Author Contributions

**Conceptualization:** Heidi Hehnly.

**Data curation:** Abrar A. Aljiboury, Eric Ingram, Nikhila Krishnan, Favour Ononiwu, Debadrita Pal, Julie Manikas, Christopher Taveras, Nicole A. Hall, Jonah Da Silva, Heidi Hehnly.

**Formal analysis:** Abrar A. Aljiboury, Eric Ingram, Nikhila Krishnan, Favour Ononiwu, Debadrita Pal, Julie Manikas, Christopher Taveras, Nicole A. Hall, Jonah Da Silva, Heidi Hehnly.

**Funding acquisition:** Heidi Hehnly.

**Investigation:** Abrar A. Aljiboury, Eric Ingram, Nikhila Krishnan, Favour Ononiwu, Debadrita Pal, Julie Manikas.

**Methodology:** Abrar A. Aljiboury, Eric Ingram, Nikhila Krishnan, Favour Ononiwu, Judy Freshour, Heidi Hehnly.

**Project administration:** Heidi Hehnly.

**Resources:** Judy Freshour, Heidi Hehnly.

**Supervision:** Heidi Hehnly.

**Validation:** Heidi Hehnly.

**Visualization:** Abrar A. Aljiboury, Eric Ingram, Nikhila Krishnan, Favour Ononiwu, Debadrita Pal, Julie Manikas, Nicole A. Hall, Heidi Hehnly.

**Writing – original draft:** Heidi Hehnly.

**Writing – review & editing:** Abrar A. Aljiboury, Heidi Hehnly.

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
