## [Decision Letter · Decision Letter 0]

10 Mar 2023

Dear Dr Hehnly,

Thank you very much for submitting your Research Article entitled 'Rab8, Rab11, and Rab35 coordinate lumen and cilia formation during Zebrafish Left-Right Organizer development' to PLOS Genetics.

The manuscript was fully evaluated at the editorial level and by independent peer reviewers. We apologize for the lengthy time that your manuscript was under review. Overall the reviewers are very positive about the studies in your manuscript, but have raised some substantial concerns. The reviewers make many valuable suggestions to improve the manuscript writing and progression in presenting the results, as well as points regarding the origin of the KV and the basis for the heart laterality looping defect that should be addressed in a revision.  Based on the reviews, we will not be able to accept this version of the manuscript, but we would be willing to review a much-revised version.

If you decide to revise the manuscript for further consideration at PLOS Genetics, please aim to resubmit within the next 60 days, unless it will take extra time to address the concerns of the reviewers, in which case we would appreciate an expected resubmission date by email to plosgenetics@plos.org.

Please do not hesitate to contact us if you have any concerns or questions.

Yours sincerely,

Mary

Mary C. Mullins

Academic Editor

PLOS Genetics

Gregory Barsh

Editor-in-Chief

PLOS Genetics

Reviewer's Responses to Questions

**Comments to the Authors:**

Reviewer #1: The manuscript by Aljiboury and co-authors examines lumen and cilia formation in Kupffer’s Vesicle (KV), the zebrafish laterality organ, with a specific focus on the roles of the small GTPases Rab8, 11 and 35. In mammalian tissue culture systems all 3 Rabs have been implicated in ciliogenesis and/or lumen formation. In this work, the authors show that Rab35 and Rab11 play key roles in lumen formation while Rab8 is not required. They show that Rab8 is important for cilia elongation. The authors also provide convincing evidence that cilia in the KV are formed inside the cell via a pathway they term the intracellular pathway. The authors use a combination of experimental approaches to produce high quality data and the findings should be of broad interest to cell and developmental biologists. Some minor suggestions are outlined below.

-page 3 line 20. The authors refer to the source of KV cells as being the endoderm. The KV comes from a subset of dorsal EVL cells (Oteiza et al., 2008; Warga and Kane, 2018). While the DFCs express both endodermal and mesodermal markers, after KV breakdown cells contribute to mesodermal structures (notochord and somites) (Melby et al., 1996). The common view in the field is that the germ layer identity of DFCs is ambiguous (Stainier and Warga 2002), so I suggest removing or qualifying statement the that the KV comes from the endoderm.

-The legends for Figures 1 and 2 are rather terse—it would be helpful to add some detail so that the reader does not have to refer to the main text to glean the major points.

-In Fig S1D the actin in black is not visible.

Reviewer #2: This manuscript has investigated the roles of three Rabs in controlling the formation of a ciliated Kupffer's vesicle, the transient structure that forms in the zebrafish tailbud where left-right asymmetry is thought to be first established. The authors use a range of immunostaining and live imaging, together with optogenetic and morpholino loss-of-function studies to demonstrate roles for Rabs in lumen expansion and proper cilia formation. They then demonstrate that the lumen expansion defect is likely due to failure of trafficking CFTR to the apical membrane. In the last section of the manuscript, they perform some left-right patterning phenotyping and find defects when Rabs are disrupted. This manuscript contains high quality data and makes some novel findings. As such, I am overall supportive of publication. However, there were two major concerns: 1) the manuscript was very difficult to read and; 2) I am not convinced the left-right heart phenotypes reflect issues at KV. I detail these concerns, plus some other more minor issues, below:

Major comments:

This was a challenging paper to read, as the writing is dense, the narrative jumps around, some critical background information is not provided and the main body of the text is not broken into defined results. For instance, the first paragraph of the results is about Rab localization but then the manuscript departs from that topic to discuss the serendipitous observation of intracellular ciliogenesis. It is not until much later that we return to Rabs and perform functional experiments.

Recommendations: 1) Reorder some of the results to walk the reader more logically through the data; 2) break the results section into subheadings to help the reader get the main point of what is described in the text in that section; 3) Add some background information on Rab GTPases and their potential roles in the Introduction. Many readers will not be automatically familiar with Rab pathways.

Rab localization studies use overexpression of tagged mRNAs. How do we know the localization we are seeing is not caused by overexpression?

The interpretation of the left-right patterning phenotype in Figure 4 is not very convincing given how developmentally abnormal some of the embryos are more broadly. How can the authors say the abnormal heart looping phenotype is caused by defective KV lumen expansion/cilia phenotypes specifically when the embryos are so damaged? Is the LPM abnormal, are the heart cells themselves abnormal? This is especially important since the prevalence of abnormal looping seems quite high (even higher than what you see when Nodal is removed) and higher than what you'd expect based on their more cellular phenotyping. Also note, it has been shown that heart looping is mostly controlled by heart-intrinsic effects and not solely by KV. They are shining blue light on embryos until 12 hpf, well after the stage cardiac precursor cells have formed and are migrating. These cells likely use Rabs for various functions. Recommendation: Either more convincingly show L-R defects (additional asymmetric organs, Nodal/Pitx2 LPM staining), and show that they originate in KV (e.g. target Rab MO's to KV specifically), or perhaps remove this section from the manuscript.

Minor comments:

Line 3-4 pg 4 "…beat in a leftward motion…" This is an imprecise. Cilia generate a directional flow but they do not 'beat leftward'. Recommendation: re-phrase.

There are many typos in the manuscript e.g. in the abstract, line 6 "to establishment" and pg 4 line 15 "…similar extent that it does…". There are many others, making for a distracted read. Recommendation: proof read and correct.

At the end of page 7 and start of page 8, the writing is confusing. Are all cilia shorter in optogenetic clustered conditions, or only those "stuck in the cell volume"? (This is explained for Rab8, but not the others). Recommendation: re-phrase/explain more thoroughly.

Are the cilia length defects a knock-on consequence of failure to expand the lumen? Or do Rab35 and Rab11 play specific roles in elongating cilia in addition to expanding the lumen? Recommendation: test/mention whether CFTR mutation causes the cilia length/extension phenotype in addition to the lumen expansion phenotype, or whether these phenotypes are separable under CFTR loss-of-function conditions.

Is the curved tail phenotype always upward curl? Motile cilia are essential for axial straightening, but usually motile cilia mutants have downward curled tails. Do the authors think the origin of the tail curl is cilia phenotypes or something else? Recommendation: describe more precisely the direction of tail curl phenotypes.

**Have all data underlying the figures and results presented in the manuscript been provided?**

Reviewer #1: Yes

Reviewer #2: Yes

PLOS authors have the option to publish the peer review history of their article (what does this mean?). If published, this will include your full peer review and any attached files.

Reviewer #1: No

Reviewer #2: No

---

## [Decision Letter · Decision Letter 1]

26 Apr 2023

Dear Dr Hehnly,

We are pleased to inform you that your manuscript entitled "Rab8, Rab11, and Rab35 coordinate lumen and cilia formation during Zebrafish Left-Right Organizer development" has been editorially accepted for publication in PLOS Genetics. Congratulations!

A reviewer also noted that several figure legends should be updated to indicate that the 'Bar' is a 'Scale bar'.

Yours sincerely,

Mary

Mary C. Mullins

Academic Editor

PLOS Genetics

Gregory Barsh

Editor-in-Chief

PLOS Genetics

Comments from the reviewers (if applicable):

Reviewer's Responses to Questions

**Comments to the Authors:**

Reviewer #1: The revised manuscript is much easier to follow and all requested changes have been made.

Reviewer #2: The authors have addressed my comments well and I am supportive of publication. I would like to congratulate the authors on a very nice study.

**Have all data underlying the figures and results presented in the manuscript been provided?**

Reviewer #1: Yes

Reviewer #2: Yes

PLOS authors have the option to publish the peer review history of their article (what does this mean?). If published, this will include your full peer review and any attached files.

Reviewer #1: No

Reviewer #2: No

**Data Deposition**

http://datadryad.org/submit?journalID=pgenetics&manu=PGENETICS-D-23-00037R1

**Press Queries**

---

## [Editor Report · Acceptance letter]

9 May 2023

PGENETICS-D-23-00037R1 

Rab8, Rab11, and Rab35 coordinate lumen and cilia formation during Zebrafish Left-Right Organizer development 

Dear Dr Hehnly, 

We are pleased to inform you that your manuscript entitled "Rab8, Rab11, and Rab35 coordinate lumen and cilia formation during Zebrafish Left-Right Organizer development" has been formally accepted for publication in PLOS Genetics! Your manuscript is now with our production department and you will be notified of the publication date in due course.

With kind regards,

Anita Estes

PLOS Genetics

On behalf of:
